# Estimating the Intra-Puparial Period of *Chrysomya nigripes* Aubertin Using Morphology and Attenuated Total Reflection Fourier Transform Infrared (ATR-FTIR) Spectroscopy

**DOI:** 10.3390/insects16050480

**Published:** 2025-05-01

**Authors:** Yi Guo, Yundi Gao, Na Chen, Xin Tang, Liangliang Li, Gengwang Hu, Jiangfeng Wang, Yu Wang

**Affiliations:** 1Department of Forensic Medicine, Soochow University, Ganjiang East Road, Suzhou 215000, China; gy932757187@163.com (Y.G.); 20224221069@stu.suda.edu (Y.G.); 2230511018@stu.suda.edu.cn (N.C.); 2130511014@stu.suda.edu.cn (X.T.); 20214221010@stu.suda.edu.cn (G.H.); jfwang@suda.edu.cn (J.W.); 2Shandong University of Political Science and Law, Jiefang East Road, Jinan 250014, China; 002325@sdupsl.edu.cn

**Keywords:** forensic entomology, puparia, intra-puparial period, spectroscopy, postmortem interval, *Chrysomya nigripes*

## Abstract

*Chrysomya nigripes* Aubertin, 1932, of the Calliphoridae family, shows later colonization and longer durations on carcasses than those of other species in this family. This species can be used to estimate the minimum postmortem interval and can be applied to highly decomposed bodies and those in the skeletonized stage. There are, however, no studies on the accurate estimation of its intra-puparial age. Here, we investigated the intra-puparial morphological changes of *C. nigripes* at seven temperatures, ranging from 16 °C to 34 °C. We also investigated the potential value of Attenuated Total Reflectance Fourier Transform Infrared (ATR-FTIR) coupled with chemometric methods for the intra-puparial age estimation of *C. nigripes* at 19 °C, 25 °C, and 31 °C.

## 1. Introduction

The minimum postmortem interval (PMI_min_) in forensic entomology refers to the time interval between the corpse colonization of sarcosaprophagous insects and the discovery of the corpse [1]. Sarcosaprophagous insects are attracted to decaying corpses by the odor, and subsequently colonize and feed on the corpse [2], allowing these insects to spend their entire growth and developmental period on the corpse or/and surrounding environment. It is accepted that the PMI_min_ of a corpse can be estimated from the insect developmental cycle, irrespective of whether the corpse was dead for a few days or a few months, as long as the oldest entomological evidence is present on the corpse [3]. Therefore, researchers tend to believe that forensic entomology is one of the most effective methods of estimating the postmortem interval (PMI) [4,5].

The larval stage and the intra-puparial period of insects can be used for PMI_min_ estimation. When using larvae as a tool for estimating PMI_min_, the indicators used usually include larval body length changes, developmental duration, and the thermal requirement of each instar. This method is more mature, and is used more often. However, the intra-puparial period cannot be aged in the same way because pupae are not accompanied by obvious changes in external morphology during development [6]. However, the intra-puparial morphology changes significantly over time, and can therefore be used as an indicator to estimate the intra-puparial period. Although the intra-puparial period is less studied compared to the larval stage, the intra-puparial developmental period of many Dipteran species accounts for more than 50% of their entire life cycle [7,8].

The intra-puparial period has been categorized into different stages that correspond to a series of events and processes that occur during metamorphosis within the pupae. In 1973, Fraenkel and Bhaskaran [9] studied the intra-puparial development of cyclorrhaphous flies (Diptera) by histological methods and established a nomenclature for the different stages of the intra-puparial development. In 1991, Greenberg [10] described 11 intra-puparial developmental periods for *Phormia regina* Meigen, 1826 and this was the first species-level forensic study to directly link developmental periods to morphological changes [11]. These criteria are now gradually being refined and applied to the study of the intra-puparial morphology of various fly species [12,13,14].

In addition to studies removing the puparium to observe morphological changes within the pupae under a stereomicroscope, research in recent years has also used a variety of imaging techniques for intra-puparial stage observation, including optical coherence tomography [15], microcomputed tomography [16,17], hyperspectral imaging [18], and scanning electron microscopy [19]. Meanwhile, several studies have established methods for estimating intra-puparial age using gene expression changes [20], cuticular hydrocarbons [21], and gas/liquid chromatography-mass spectrometry [22]. Although these methods provide new insights and methods to estimate intra-puparial age, most require complex equipment and expensive reagents and a large number of samples [23], so the search for a cheaper, faster, and more convenient method to estimate intra-puparial age is still ongoing.

Attenuated Total Reflectance Fourier Transform Infrared (ATR-FTIR) is an analytical tool for analyzing the structure and chemical composition of compounds based on the vibrational modes of molecular functional groups, thus providing analytical fingerprinting [24]. This analytical method is low cost, sample preparation is simple and retains the original appearance of the sample for morphology, is more suitable for observing changes on the sample surface, and can be used for qualitative and quantitative analysis of the sample [25]. Currently, ATR-FTIR combined with chemometric methods is widely used in forensic science, often for the identification of body fluids [26], hair [27], soil [28], and myocardial fibrosis [29].

The combination of ATR-FTIR and chemometrics provides a powerful tool for chemical analysis, enabling high-quality quantitative and qualitative results to be obtained alongside rapid, non-destructive analysis. The application of this method is promising, especially in research areas where high throughput and sensitivity are required. However, compared to the use of ATR-FTIR in other areas of forensic science, the application of ATR-FTIR in forensic entomology, especially for the aging of sarcosaprophagous insects, is rare [23,30]. *Chrysomya nigripes* is a species of Calliphoridae, but unlike other species of this family, it arrives later and is present for a longer period on the corpse, making it valuable in estimating a longer PMI. Studies have been conducted on its developmental duration and indicators of larval development [31], but there are no detailed studies on the intra-puparial period of *C. nigripes*. Therefore, research on the indicators of change in the intra-puparial period of this species will further improve the accuracy of PMI_min_ estimation using this species. The aim of this study was to investigate the morphological changes of the pupae of *C. nigripes* and the value of using a combination of ATR-FTIR and chemometrics for the estimation of the intra-puparial age.

## 2. Materials and Methods

### 2.1. Morphological Study

Laboratory population establishment of *C. nigripes* was carried out according to Guo et al. [31]. The wandering larvae were collected in Zhongshan, China (22°30′ N, 113°23′ E), and transported to the forensic entomology laboratory of Soochow University for the establishment of a laboratory population. The larvae were reared to adults in an intelligent light incubator at 25 °C. After eclosion, the adults were maintained in a rearing cage measuring 50 × 50 × 50 cm, where an equal mixture of powdered milk and white sugar was provided to promote sexual maturation of the adults. Following the maturation of the adults, 40 g of decomposed pork was placed in a beaker covered with a layer of pig skin, and the beaker was placed in a rearing cage to attract egg-laying. The eggs were observed every hour, and after laying, the eggs were transferred to an intelligent light incubator. This procedure was repeated until the population grew to include around 2500 adults in each rearing cage.

About 300 eggs laid by adult insects within two hours were transferred to petri dishes with rotten pork and placed in insect rearing boxes with 2 cm of vermiculite at the bottom. Each petri dish was placed separately in seven intelligent light incubators KXG-300 (Yingmin Co., Ltd., Changzhou, China) at 16 °C, 19 °C, 22 °C, 25 °C, 28 °C, 31 °C, and 34 °C, with a humidity of 70% and a photoperiod of L12:D12. Observations were made once every 8 h, and rotting pork was added according to the rate of consumption. *Chrysomya nigripes* could not complete its development at 16 °C, so the experiment was carried out at the other six constant temperatures. When about 50% of the wandering larvae formed white pre-pupae, five pupae were sampled every 8 h until eclosion at each temperature. The samples were killed with hot water (>90 °C) and placed in 80% ethanol for storage. The process was repeated three times at each temperature. Then, the puparium was carefully removed with an insect needle and tweezers, and the intra-puparial morphology was observed, recorded, and photographed under a Nexcope NSZ818 research-grade compound achromatic parallel-light stereomicroscope. The external morphology of the intra-puparial stage of *C. nigripes* was divided into 12 sub-stages (A–L) based on previous studies [32,33].

### 2.2. Attenuated Total Reflectance Fourier Transform Infrared (ATR-FTIR)

The pupae for ATR-FTIR detection were reared at 19 °C, 25 °C, and 31 °C in the same way as the samples for morphological analysis. Five pupae were taken from each temperature at 24-h intervals until eclosion, and the last samples were five empty puparia. After sampling, the surface of the pupae/puparia was cleaned with an ultrasonic cleaner (Fan Ying Technology Co., Ltd., Zhongshan, China). The dorsal puparium at the second thoracic segment (ca. 1.1 × 2.2 mm) of each pupa was removed with forceps and scissors. Subsequently, all puparia pieces were placed on a dry piece of paper towel to remove any excess moisture. After that, the puparium was left to air-dry in a well-ventilated environment for 1 h. Once thoroughly dried, the samples were then carefully placed in a 1.5 mL EP tube and stored at −80 °C for further processing. All the samples stored at −80 °C underwent ATR-FTIR detection within a week.

All spectra were obtained using a Nicolet-iS 5 FTIR spectrometer (Thermo Electron Scientific Instruments Corp, Madison, WI, USA) equipped with an iD7 ATR accessory following the protocols of Barbosa et al. [23] and Oliveira et al. [34]. Spectroscopic measurements were performed in an acclimatized room, controlling the temperature (22 °C) and relative air humidity (60%), and allowing the samples to stabilize at that temperature before analysis. The detection window of the ATR attachment was cleaned with anhydrous ethanol before each measurement, and a blank background spectrum was taken before sampling. During measurement, the sample was placed directly onto the ATR attachment. The parameters for spectral acquisition were set to a frequency of 4000–600 cm^−1^ with a resolution of 4 cm^−1^ and 32 scans. Five measurements were obtained for each sampling time point at each constant temperature (19 °C, 25 °C, 31 °C), and these five spectra were combined to obtain an averaged spectrum representing the sample to minimize the error. Spectral data were recorded using the infra spectra analysis software package OMNIC version 8.2. The fingerprint region for further analysis was 1800–900 cm^−1^, and smoothing denoising preprocessing (Savitzky-Golay convolution algorithm, smoothing point 5), standard normal variate (SNV), baseline correction for SNV normalized spectral data, and mean center correction were performed on all selected spectral regions to eliminate spectral overlap (Figure 1) [35].

### 2.3. Analysis of Chemometric Methods

The spectral data of puparium samples at different temperatures at different time points were downscaled using a principal component analysis (PCA) in order to improve the predictive capability of the machine-learning-based prediction models. The original feature set was transformed into a dimensionality reduction using PCA to extract the relevant variables from the original data, transform them into a smaller set of uncorrelated variables, and reveal the similarities and differences in the data through a score plot using the sample projections on the given PC [36]. The high-dimensional data was converted to low-latitude data by PCA to reduce computational effort, followed by projecting the data into a two-dimensional space for easy data visualization. The raw data were then divided into a training set (60%) and a validation set (40%). The training set was used to train the classifier, and then the validation set was employed to test the model obtained from the training.

Partial Least Squares Discriminant Analysis (PLS-DA) and Random Forest (RF) are two machine learning (supervised) models that can be used for classification and regression problems. After dimensionality reduction by the PCA model, the reduced low-dimensional data were applied to the PLS-DA and RF models for validation. The spectral data were used to build the PLS-DA and RF test set/training confusion matrix for the estimation of the *C. nigripes* intra-puparial periods, and the results predicted by the model were compared with the actual observed results, and the number of correct and incorrect classification results were recorded. Therefore, the model’s Accuracy (A), Recall (R), Precision (P), and F1-Score metrics were computed to evaluate the model’s performance. Validation was performed three times at each temperature to ensure that as much data as possible were trained, and the test set/training set classification results were compared to calculate the average accuracy, precision, recall, and F1 score. The specific formulae are as follows:$$\mathrm{Accuracy}=\frac{TP+TN}{TP+TN+FP+FN}$$$$\mathrm{Precision}=\frac{TP}{TP+FP}$$$$\mathrm{Recall}=\frac{TP}{TP+FN}$$$$F1-\mathrm{Score}=\frac{2\times precision\times recall}{precision+recall}$$$$\mathrm{Macro}-\mathrm{Accuracy}=\frac{A1+A2+A3+\ldots +An}{n}$$$$\mathrm{Macro}-\mathrm{Precision}=\frac{P1+P2+P3+\ldots +Pn}{n}$$$$\mathrm{Macro}-\mathrm{Recall}=\frac{R1+R2+R3+\ldots +Rn}{n}$$
where True Positive (TP) is the number for which the true value is positive and which the model considers positive; False Negative (FN) is the number for which the true value is positive and which the model considers negative; False Positive (FP) is the number for which the true value is negative and which the model considers positive; True Negative (TN) is the number for which the true value is negative and the model considers negative.

All data were analyzed using OMNIC version 8.2 (Thermo Nicolet Analytical Instruments, Madison, WI, USA), origin 2019b, and MATLAB R2020b (The MathWorks, Natick, MA, USA).

## 3. Results and Discussion

### 3.1. Morphological Observations

#### 3.1.1. Overall Morphological Changes Within the Pupae

*Chrysomya nigripes* could not complete its developmental process at 16 °C but completed the entire development of the intra-puparial period at the six constant temperatures ranging from 19 °C to 34 °C, with the duration of the intra-puparial period ranging from a maximum of 192 ± 0 h at 19 °C to 77.3 ± 4.6 h at 34 °C. Morphological changes within the intra-puparial period were divided into 12 sub-stages (A–L), and the time required for each stage at the different temperatures is shown in Appendix A and Figure 2. The typical characteristics of each stage are described below.

A: Pre-pupal stage

The internal tissues are difficult to separate from the puparium; larval-pupal apolysis was incomplete. The pupa was easily broken during dissection and had a rough, yellowish-white surface (Figure 3(A1–A3)).

B: Early-cryptocephalic pupal stage

Internal tissues and puparium can be separated but are still easily broken; surface rough, yellowish-white; legs and wings shorter than 1/3 of the body; compound eyes not visible; the respiratory horns darkened in color and located in the anterior part of the pupae (Figure 3(B1–B3)).

C: Late-cryptocephalic pupal stage

The cryptocephalic pupa is yellowish white with a smooth surface; the head, thorax, and abdomen have larval-like segments; legs and wings are less than 1/2 the length of the body, clinging to the body (Figure 3(C1–C3)).

D: Phanerocephalic pupal stage

The phanerocephalic pupa is yellowish white with a transparent yet visible pupal cuticle attached to the pupa. The head, thorax, and abdomen are segmented but without clear demarcation lines; the mouthparts appear square-shaped; the legs and wings are thick, and they are all more than half the length of the body (Figure 3(D1–D3)).

E: Pharate adult stage I

With the completion of pupal-adult apolysis, the cuticle of the pupa separates from the surface of the pupa, and the pupa becomes a pharate adult. The color of the pupa is yellowish white and clear boundaries can be seen between the head and thorax, as well as between the thorax and abdomen; the antennae are differentiated but not yet fully developed; the legs and wings are thinned; the mouthparts appear bifid (Figure 3(E1–E3)).

F: Pharate adult stage II

The pharate adult is yellowish white; the antennae are distinctly outlined; mouthparts are elongated and narrowed but had not reached their full length; labial palps appear on the upper lip; legs and wings are thin; wing veins are visible and partially folded (Figure 3(F1–F3)).

G: Pharate adult stage III

The pharate adult is yellowish white. The antennae and mouthparts are clearly visible and both have reached their full length; the thoracic dorsal bristles are beginning to appear; the abdomen is distinctly segmented; wings fully folded (Figure 3(G1–G3)).

H: Pharate adult stage IV

The pharate adult is yellowish white. The compound eyes are pinkish red and the legs light brown (Figure 3(H1–H3)).

I: Pharate adult stage V

The pharate adult is light brown. Compound eyes purplish red; thoracic bristles present; abdominal setae light brown; wing margins pigmented and light brown; upper lip light brown; legs tan (Figure 3(I1–I3)).

J: Pharate adult stage VI

The color of the pharate adult is light gray; antennae and mouthparts are brownish-black; wings are light gray with black margins; the legs are black; setae are black all over (Figure 3(J1–J3)).

K: Pharate adult stage VII

The pharate adult is darker in color; compound are eyes brownish red; wings are dark gray; legs are black; antennae are grayish black; mouthparts are grayish black (Figure 3(K1–K3)).

L: Pharate adult stage VIII

The pharate adult is grayish black; compound eyes are brownish; wings are black; mouthparts are black (Figure 3(L1–L3)).

Developmental data on the intra-puparial period of *C. nigripes* have only been studied by Li et al. [37], O’Flynn [38], and Guo et al. [31], where these authors reported similar findings to the present study. Li studied the developmental period of *C. nigripes* at 20 °C, 24 °C, 28 °C, and 32 °C, reporting intra-puparial periods of 168.0 h, 112.0 h, 80.0 h, and 77.0 h, respectively. O’Flynn only studied the intra-puparial period for about 4–5 d at 28 °C, while the intra-puparial stages of *C. nigripes* were shortened from 188.8 h to 74.2 h at six temperatures from 19 °C to 34 °C in Guo’s study. Unlike other Calliphoridae species, the duration of the intra-puparial stage of *C. nigripes* takes up less than 40% of the entire immature stage, while in *C. megacephala* [7] and *C. rufifacies* [8], it can take up to 50–70% of the entire immature stage.

Similar to other species of Diptera, the development within the puparium of the *C. nigripes* goes through the pre-pupal, cryptocephalic pupal, phanerocephalic pupal, and pharate adult stages. However, the entire process of intra-puparial development is not linear. It is rapid in the early stages (e.g., stages A–C in Figure 2), slow during the middle stages (stages D–H in Figure 2), and becomes rapid again in the later stages (stages I–L in Figure 2). As a result, a distinct ‘bubble’ is formed on each graph in Figure 2. In fact, this represents a noticeable limitation of this method. In actual cases, the investigators can only use the minimum value as a reference for PMI_min_. This is precisely why many current studies, including the present study, combine multiple methods for estimating the intra-puparial age.

Similar to previous studies, the present study performs a more detailed stage division for the cryptocephalic pupal and pharate adult stages. We found that in the cryptocephalic pupal stage, although the head was not fully differentiated, the legs and wings were already differentiated. Based on the relative length of the legs and wings compared to the total length of the pupa, this stage can be subdivided into the early and late cryptocephalic pupal stages. The pharate adult stages last the longest, accounting for more than 65% of the entire intra-puparial period. Pujol-Luz and Barros-Cordeiro [39] classified the intra-pupal morphological changes of *Chrysomya albiceps* into nine stages, and the pharate adult stage into four sub-stages based on the color changes of the compound eyes. Shuvra and Santanu [19] also classified the pharate adult stage of *Sarcophaga dux* into five sub-stages based on compound eye color. Here, the pharate adult stage was divided into eight sub-stages by comprehensively analyzing the morphological and color changes of key structures, including the compound eyes, legs, wings, antennae, and mouthparts.

#### 3.1.2. Morphological Variations Within the Pupa: Compound Eyes, Thorax, and Abdomen

The overall intra-puparial morphology of *C. nigripes* is described above, but its details are not always clearly recognizable, especially when the insect evidence collected at the scene of the crime is incomplete. This is why clear, detailed structural drawings are needed for intra-puparial age estimation. Therefore, we studied the morphological characteristics of the compound eyes, antennae, mouthparts, thorax, legs, wings, and abdomen over time; the typical characteristics of the parts of each stage are described below, and the minimum and maximum time required for each stage at different temperatures are listed (Figure 4, Figure 5, Figure 6, Figure 7, Figure 8, Figure 9 and Figure 10) (Table 1, Table 2 and Table 3). The development of the compound eyes and abdomen is divided into six sub-stages. The development of the thorax and mouthparts is divided into seven sub-stages, in addition to eight sub-stages for the development of the antennae, wings, and legs.

When the wandering larvae irreversibly shorten and form a barrel-shaped puparium, the tissues within the puparium that belonged to the larvae begin to dissociate, differentiate, and reorganize, resulting in the gradual formation of the pre-pupa, cryptocephalic pupa, phanerocephalic pupa, and pharate adult, which are structurally and functionally completely different from the larvae. During this process, the compound eyes, antennae, mouthparts, thorax, legs, wings, and abdomen undergo changes, and subsequently, the organs of the pharate adult gradually develop fully and complete coloration, eventually breaking through the puparium, completing its development in the intra-puparial stage. In this study, the developmental process of each structure or organ was divided into six to eight stages based on the presence or absence of these parts or organs, their degree of development, and coloration. The methodology of this study is consistent with that of the methods of Brown et al. [40], Wang et al. [32,41], and Li et al. [42], and the changes observed in various characteristic organs within the puparium of *C. nigripes* were similar to those of *Calliphora vicina*, *Lucilia illustris*, *Calliphora grahami*, and *Sarcophaga peregrina*. The above results suggest that the subdivision of morphological characteristics of the whole, parts, and organs within the puparium over time provides a reliable basis for age estimation during the intra-puparial stage.

### 3.2. ATR-FTIR Variation over the Intra-Puparial Period

The 1800–900 cm^−1^ spectral range selected in this study provides maximum information about the compounds present in the samples, including lipid esters (1800–1700 cm^−1^); amides I, II, and III (1700–500 cm^−1^, 1350–1200 cm^−1^), and nucleic acids and carbohydrates (1200–900 cm^−1^) [43,44]. We focused on some of these spectrally characterized peaks, which are shown in Table 4. Figure 11 shows the average ATR-FTIR spectra of *C. nigripes* puparia at different constant temperatures for each intra-puparial age (day) in the range of 1800–900 cm^−1^. Several intense absorption bands were observed in the 1800–900 cm^−1^ range, and two distinct peaks were associated with proteins between 1700 and 1500 cm^−1^, with amide I and amide II present near 1632 cm^−1^ and 1516 cm^−1^, respectively. The range of 1310–900 cm^−1^ showed overlapping bands associated with C-O, C-O-H, and P-O vibrations enriched in carbohydrates, phosphates, DNA, and RNA [45]. Pickering et al. [46] studied the ATR-FTIR spectra of three Diptera species at various stages of development and also reported the presence of amide I and amide II at 1679 cm^−1^ and 1561 cm^−1^. After analyzing these major components, it was concluded that the spectral differences between the second- and third-instar larvae were mainly due to the presence of amide I and amide II in the protein. The amide functional group is part of the amino acid structure, and in addition to being present in the cuticle, it has been used to identify sarcosaprophagous flies of forensic importance by analyzing the chemical characteristics of eggs [47].

The 192 h and 216 h average spectra at 19 °C are separated from the other time spectra; this is because of the long developmental duration of this species at 19 °C, with eclosion ranging from 192 to 216 h. The samples may undergo chemical reactions (e.g., oxidation, degradation, polymerization) over a long period of time (192 h, 216 h) to produce substances containing strong absorptive functional groups (e.g., hydroxyl -OH, carbonyl C=O). These functional groups increase the absorption of infrared light at specific wavelengths, resulting in a significant decrease in reflectance. The average spectra at 0 d for the three temperatures were almost the same, with a distinct absorption peak at 1395 cm^−1^ at 0 d, which was due to the fibrinogen/methyl bending of amino acid side chains, lipids, and proteins [48]. This absorption peak weakened with time, probably due to the decomposition of proteins, lipids, and other chemicals, such as fibrinogen in the puparium, which is consistent with the findings of Li et al. [49]. de Paula et al. [50] reported that a carboxylic acid peak appeared at 1241 cm^−1^ in the epidermal samples of *C. megacephala*, which was also observed in the epidermal samples of ants by Tofolo et al. [51] and Bernardi et al. [52]. Because lipids present in insect epidermis are thought to play a key role in waterproofing and chemical communication, these components undergo regular changes during pupal development and can be used to age pupae. Most of the current ATR-FTIR studies on the intra-puparial period choose pupal tissue after liquid nitrogen grinding as the measurement sample [30,53], while in this study, we used the puparium as the sample for ATR-FTIR spectroscopy measurement and found that this method is easier, the steps are simple, and the effect of moisture on the measurement results can be excluded, especially when the number of pupae is small. Also, the puparium can be removed for the ATR-FTIR, and the remaining pupal tissues can be used for morphological observations, saving samples from being wasted.

### 3.3. Chemometric Results

The results of the PCA are shown in Figure 12. From the infrared spectral score plots of the samples, it can be seen that, at different constant development temperatures (19 °C, 25 °C, and 31 °C), PC1 explains 55.00%, 52.53%, and 53.03% of the centralized variance, and PC2 explains 27.09%, 28.68%, and 15.31% of the centralized variance, respectively. The interpretation of the variation of the score plots of the spectral data at different times (d) on PC1 and PC2 were 82.09%, 81.21%, and 68.34%. Although the samples at different time points at different temperatures had intergroup reproducibility and intergroup differentiation, there were still intersections. The longer the time intervals, the more pronounced the distinction between the groups. There was a clear separation trend between the 0–48 h and 192–216 h groups at 19 °C, 0–48 h and 96–144 h at 25 °C. Therefore, we combined PLS-DA and RF to further analyze these data.

Two classifiers, PLS-DA and RF, were model-trained based on the same dataset using the confusion matrix (Figure 13 and Figure 14), and finally, a multi-metric comparison of the two classification models was performed to obtain Table 5, where the closer the accuracy is to 100% and the closer the recall, precision, and F1 scores are to 1, the better the model’s output is represented. Therefore, for the data in this study, the PLS-DA classification model had higher reliability and accuracy and could be used to classify the puparium samples from this work.

In chemometrics, selecting an appropriate machine learning model is pivotal for ensuring the accuracy and reliability of data analysis. The PLS-DA employed in this study is well-suited for handling high-dimensional data, particularly in spectral data analysis [54,55,56]. It can elucidate the relationship between variables and classification results, which is essential for understanding the connection between components and their spectra. Conversely, RF can evaluate the contribution of individual features to classification outcomes, aiding in identifying the key variables in the analysis. Although many other machine learning models, such as support vector machines and neural networks, demonstrate good performance in certain scenarios, they generally demand more parameter tuning and intricate model design. These complex models also require longer training and prediction times. In contrast, PLS-DA and RF are computationally more efficient, making them more appropriate for rapid analysis and real-time applications in this study. However, future research should still assess the data prediction accuracy of different models to ensure more precise PMI_min_.

## 4. Conclusions

The intra-puparial period accounts for 50% of the entire immature stage, which is of great significance for the estimation of PMI_min_. This study proposes that using ATR-FTIR in combination with chemometrics serves as a supplementary tool for morphological aging assessment. There are two main advantages to combining these two methods. Firstly, calculate intra-puparial age separately with each method, and then use the intersection of the results from the two methods to further narrow down the PMI_min_ time window. This way, we can take full advantage of the unique aspects of each method. Secondly, there are situations where the ATR-FTIR device may not be easily accessible. In such cases, investigators can estimate PMI_min_ by observing the internal structure within the puparium. Also, when peeling the puparium, inexperienced investigators might potentially damage the internal structure of fly pupae. In those circumstances, the combination of ATR-FTIR and chemometrics can be used to make up for potential losses of morphological data. Overall, this study presents a novel approach for estimating the age of forensically important *C. nigripes* pupae, and consequently, it may help in accurately estimating the PMI_min_ for forensic investigations.

## Figures and Tables

**Figure 1 insects-16-00480-f001:**
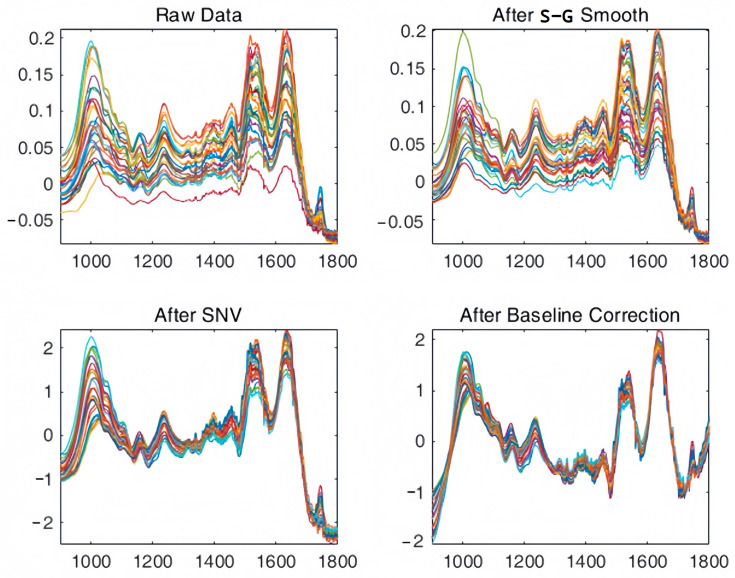
Pre-processing procedure for ATR-FTIR spectral data on the *C. nigripes* intra-puparial period, including raw data, S-G smoothing, standard normal variable (SNV), and baseline correction, using 25 °C as an example.

**Figure 2 insects-16-00480-f002:**
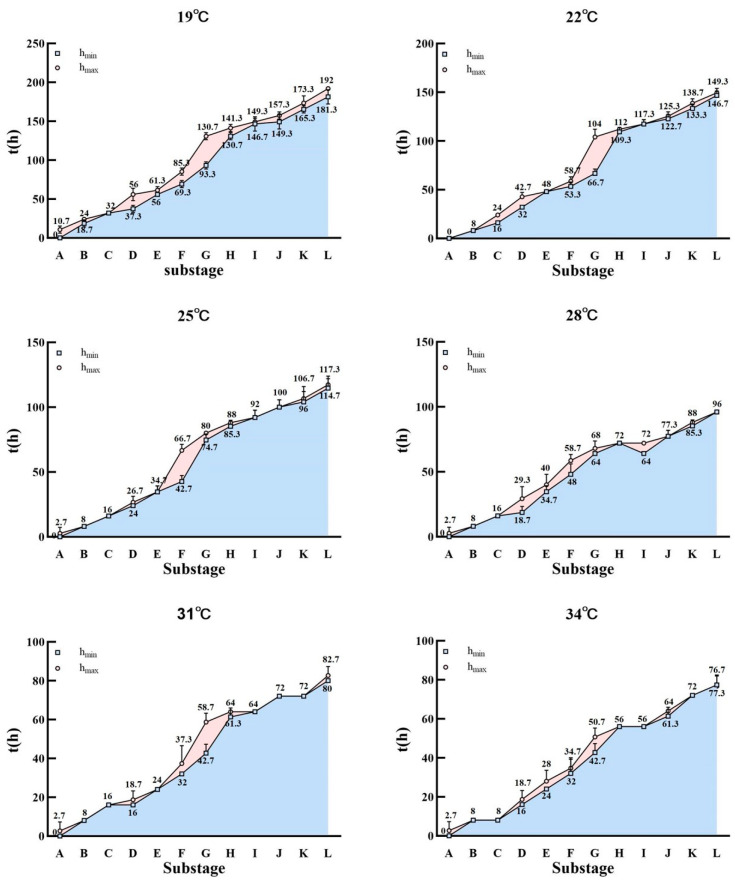
Minimum and maximum time required for each intra-puparial stage of *C. nigripes* at different temperatures. The blue square is the minimum time required for each stage; the red circle is the maximum time required for each stage; the lines are error bars.

**Figure 3 insects-16-00480-f003:**
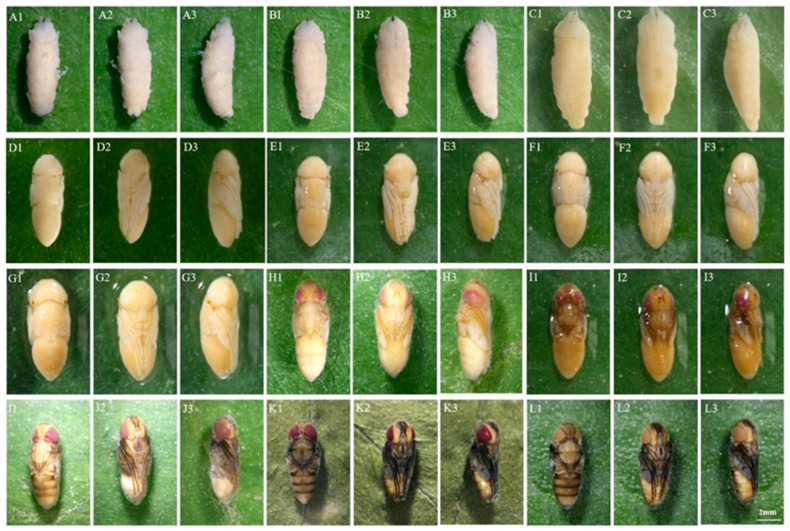
Morphological changes within the 12 pupal sub-stages (**A**–**L**) of *C. nigripes*: 1. Dorsal view. 2. Ventral view. 3. Lateral view.

**Figure 4 insects-16-00480-f004:**
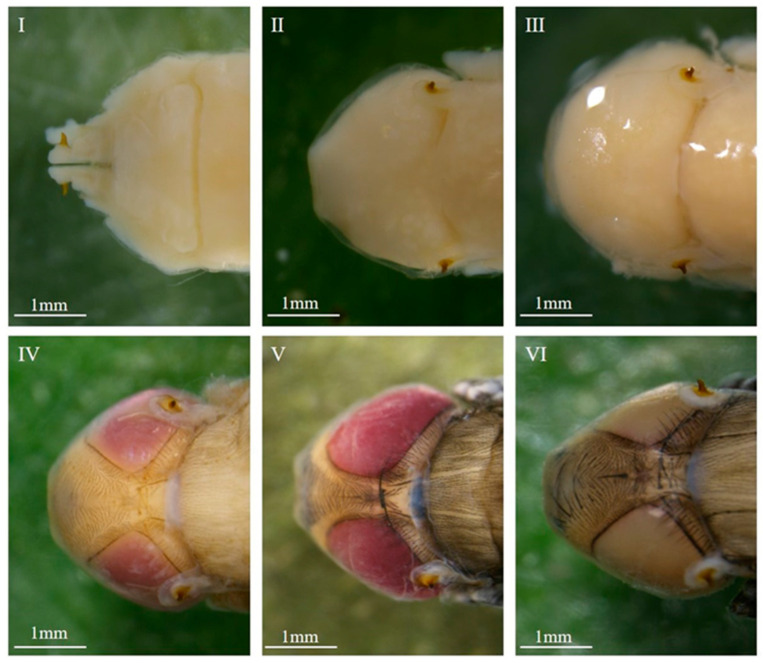
Intra-puparial development of the compound eyes of *C. nigripes*. ((**I**) se) initially there are no compound eyes; ((**II**) se) the compound eyes are fuzzy, and there is no boundary or no obvious boundary between the compound eyes and the thorax; ((**III**) se) the compound eyes are obvious, and the cephalothorax boundary is clearly defined; ((**IV**) se) pink compound eyes; ((**V**) se) purplish-red compound eyes; ((**VI**) se) brown compound eyes. “se” denotes sub-stage.

**Figure 5 insects-16-00480-f005:**
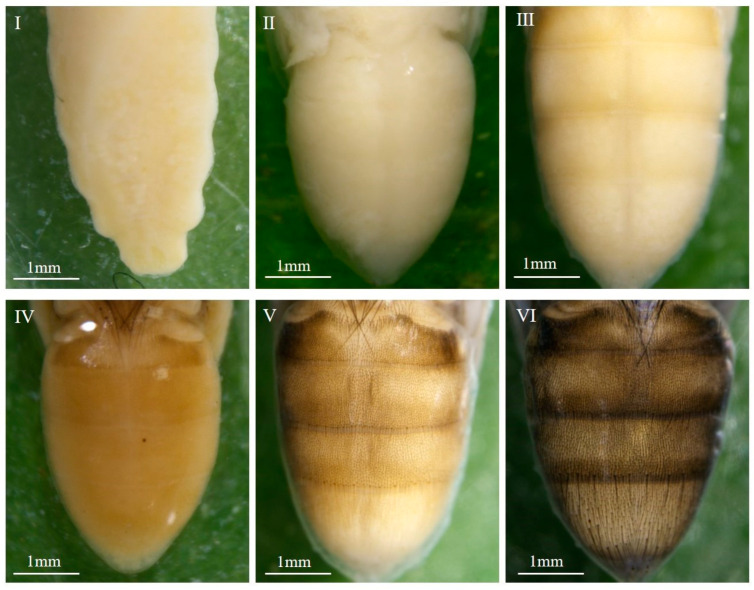
Intra-puparial development of the abdomen of *C. nigripes*. ((**I**) se) abdomen and thorax are not bound or are not clearly bound; ((**II**) se) abdomen and thorax are clearly bound; ((**III**) se) abdominal segmentation is visible; ((**IV**) se) abdomen is yellowish white; ((**V**) se) abdomen has light brown or brown hairs; ((**VI**) se) abdomen has gray-black or black hairs. “se” denotes sub-stage.

**Figure 6 insects-16-00480-f006:**
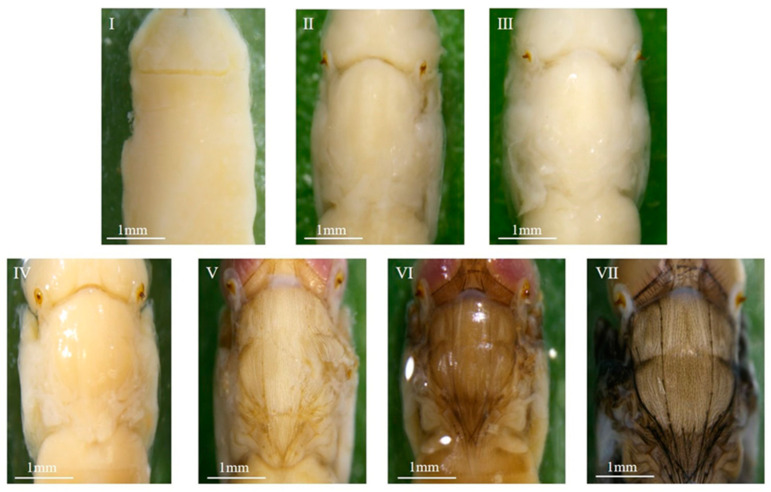
Intra-puparial development of the thorax of *C. nigripes*. ((**I**) se) no boundary or no clear boundary between the abdomen and head; ((**II**) se) clear boundary between the thorax, head, and abdomen; ((**III**) se) white setae appear on the thorax; ((**IV**) se) yellowish white thorax; ((**V**) se) brownish yellow thorax with brownish yellow dorsal seta; ((**VI**) se) brownish yellow thorax with brownish dorsal seta; ((**VII**) se) grayish black thorax with grayish black or black dorsal seta. “se” denotes sub-stage.

**Figure 7 insects-16-00480-f007:**
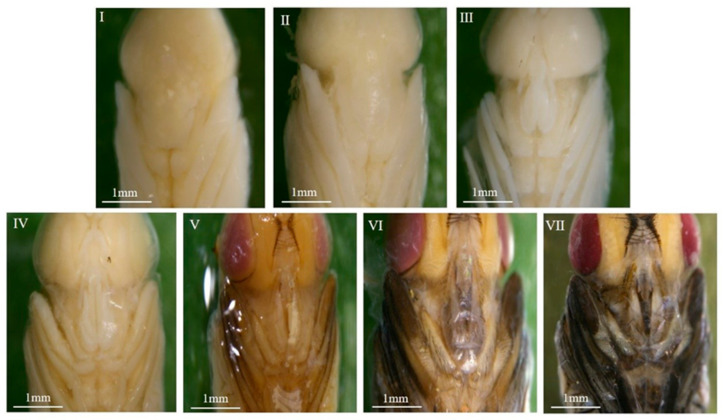
Intra-puparial development of the mouthparts of *C. nigripes*. ((**I**) se) mouthparts are a mass of undifferentiated tissue; ((**II**) se) labial flap is slightly split; ((**III**) se) labial flap and labial palps are emerging, and mouthparts are elongating but have not reached their full length; ((**IV**) se) mouthparts have developed to their full length; ((**V**) se) labial flap and labial palps are light brown; ((**VI**) se) labial flap and labial palps are light gray; ((**VII**) se) labial flap and labial palps are grayish black or black. “se” denotes sub-stage.

**Figure 8 insects-16-00480-f008:**
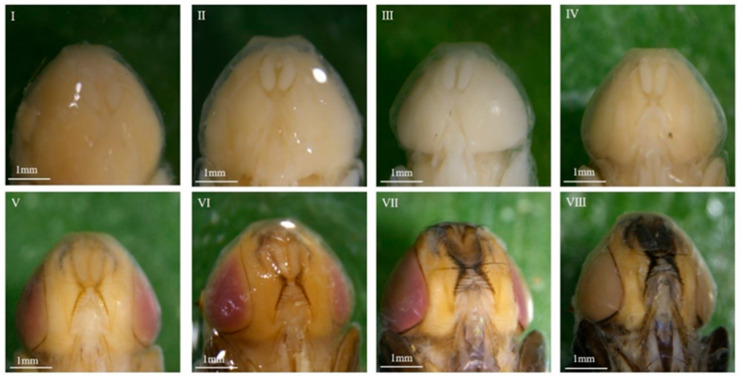
Intra-puparial development of the antennae of *C. nigripes*. ((**I**) se) antennae emerge with faint outlines; ((**II**) se) antennae fully developed with distinct outlines; ((**III**) se) antennae white and elongated, developed to full length; ((**IV**) se) antennae yellowish white; ((**V**) se) antennal margins pigmented, light brown; ((**VI**) se) antennules light brown with brown margins; ((**VII**) se) antennae dark brown with gray margins; ((**VIII**) se) antennae black with black margins. “se” denotes sub-stage.

**Figure 9 insects-16-00480-f009:**
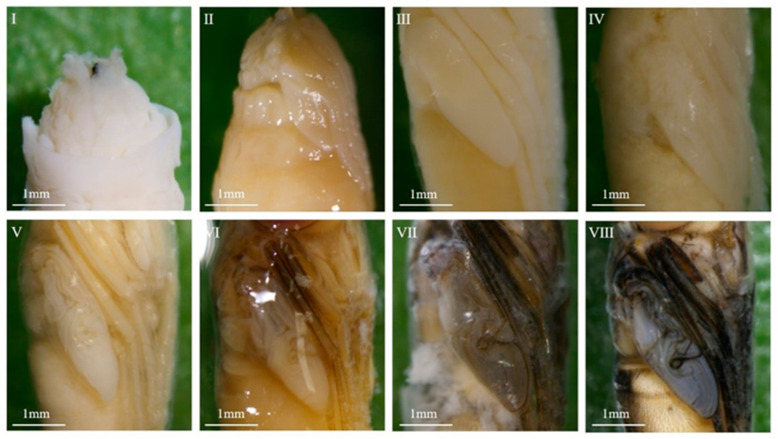
Intra-puparial development of the wings of *C. nigripes*. ((**I**) se) wings short and small, shorter than one-third of the pupal length; ((**II**) se) wings tightly attached to the body and half the length of the pupa; ((**III**) se) thick wings; ((**IV**) se) thin wings with conspicuous veins; ((**V**) se) wings folded; ((**VI**) se) wings light brown with brown pigmented margins; ((**VII**) se) wings grayish black; ((**VIII**) se) wings black. “se” denotes sub-stage.

**Figure 10 insects-16-00480-f010:**
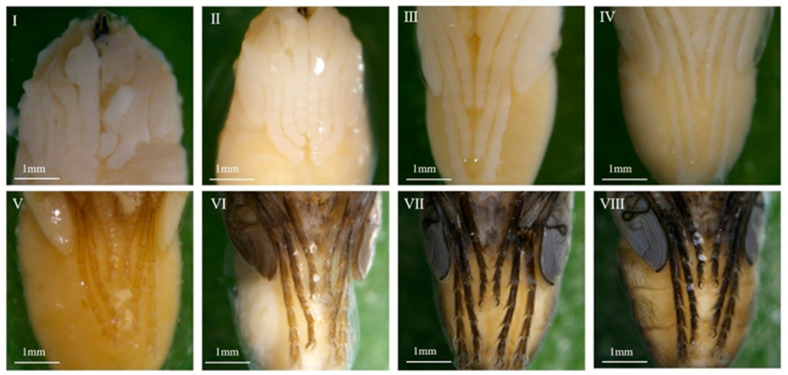
Intra-puparial development of the legs of *C. nigripes*. ((**I**) se) legs short, less than one-third of pupa length; ((**II**) se) legs elongated and tightly attached to the body, reaching half the length of the pupa; ((**III**) se) legs reached full length and thick; ((**IV**) se) legs slender, tarsus visible; ((**V**) se) legs light brown with brown pigmented margins; ((**VI**) se) legs gray; ((**VII**) se) legs dark gray; ((**VIII**) se) legs dark black. “se” denotes sub-stage.

**Figure 11 insects-16-00480-f011:**
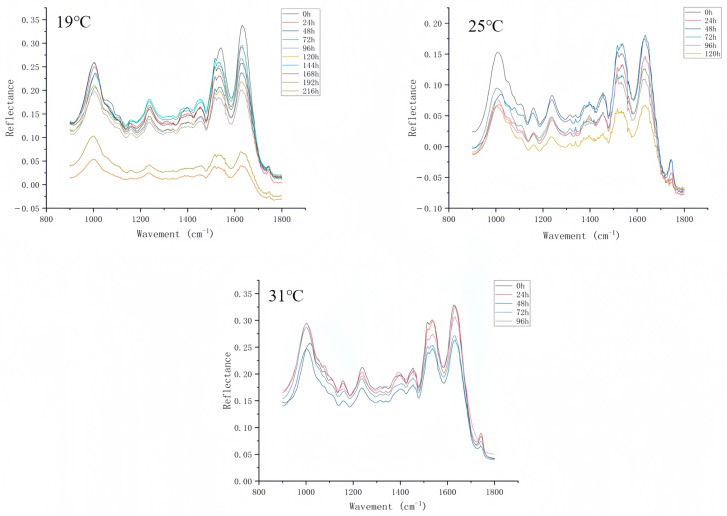
Mean ATR-FTIR spectra of *C. nigripes* puparium at different ages (days) at different constant temperatures (19 °C, 25 °C, 31 °C).

**Figure 12 insects-16-00480-f012:**
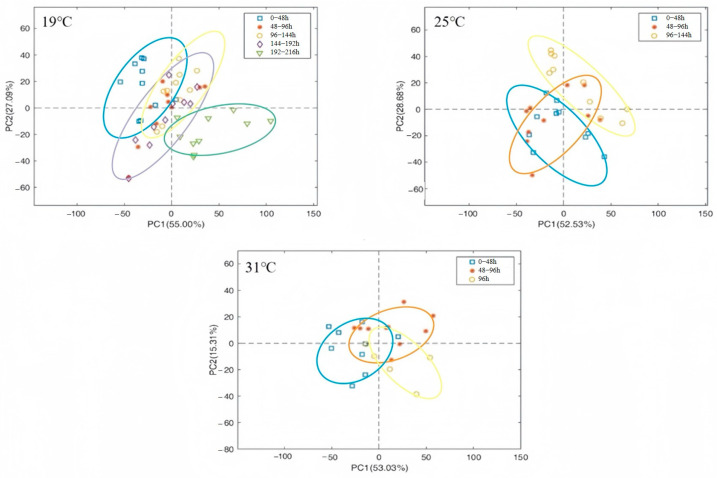
Plot of the principal component analysis scores for all spectral data of *C. nigripes* puparium at three different constant developmental temperatures (19 °C, 25 °C, 31 °C).

**Figure 13 insects-16-00480-f013:**
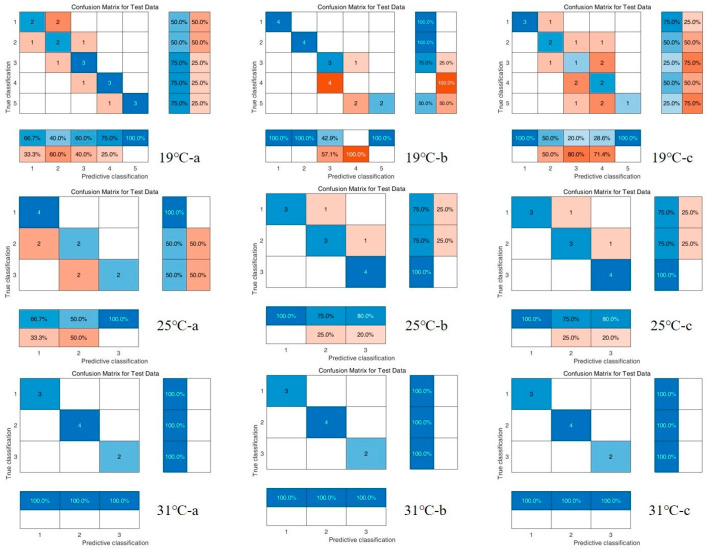
PLS-DA test set/training confusion matrix for all spectral data of *C. nigripes* puparium at three different constant developmental temperatures (19 °C, 25 °C, and 31 °C. a, b, c represents three repetitions).

**Figure 14 insects-16-00480-f014:**
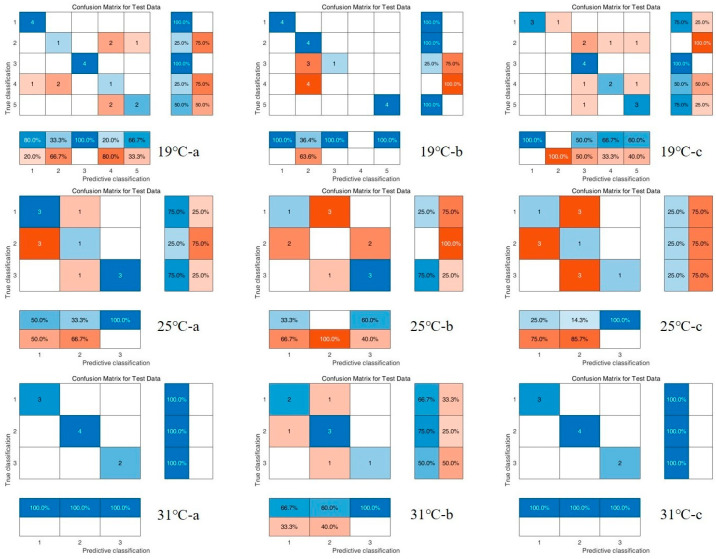
RF test set/training confusion matrix for all spectral data of *C. nigripes* puparium at three different constant developmental temperatures (19 °C, 25 °C, and 31 °C. a, b, c represents three repetitions).

**Table 1 insects-16-00480-t001:** Minimum and maximum duration of morphological changes in the compound eyes and abdomen of *C. nigripes* after pupariation at different temperatures.

Temperature		Compound Eyes (h)						Abdomen (h)					
		I	II	III	IV	V	VI	I	II	III	IV	V	VI
19 °C	h_min_	0	37.3	56	130.7	144	176	0	56	93.3	122.7	144	176
	SD	0	4.6	0	4.6	8	0	0	0	4.6	9.2	0	0
	h_max_	29.3	56	130.7	141.3	170.7	192	56	85.3	120	144	168	192
	SD	4.6	8	4.6	4.6	4.6	0	8	4.6	13.9	0	0	0
22 °C	h_min_	0	32	45.3	109.3	117.3	138.7	0	48	66.7	93.3	120	136
	SD	0	0	4.6	4.6	4.6	4.6	0	0	4.6	16.7	0	8
	h_max_	24	37.3	104	112	130.7	149.3	42.7	58.7	90.7	114.7	128	149.3
	SD	0	4.6	8	0	4.6	4.6	4.6	4.6	12.2	4.6	8	4.6
25 °C	h_min_	0	24	32	85.3	96	109.3	0	34.7	74.7	88	96	109.3
	SD	0	0	0	4.6	8	4.6	0	4.6	4.6	8	0	12.2
	h_max_	16	24	80	93.3	104	117.3	26.7	66.7	85.3	90.7	108	117.3
	SD	0	0	0	9.2	8	4.6	4.6	4.6	4.6	4.6	5.7	4.6
28 °C	h_min_	0	18.7	29.3	69.3	77.3	88	0	34.7	58.7	64	77.3	88
	SD	0	4.6	4.6	4.6	4.6	0	0	4.6	9.2	8	4.6	0
	h_max_	10.7	21.3	66.7	72	85.3	96	29.3	50.7	61.3	72	80	96
	SD	4.6	4.6	4.6	8	4.6	0	9.2	9.2	12.2	0	0	0
31 °C	h_min_	0	16	24	61.3	72	80	0	24	42.7	58.7	72	80
	SD	0	0	0	4.6	0	0	0	0	4.6	4.6	0	0
	h_max_	16	18.7	58.7	64	72	85.3	18.7	37.3	53.3	64	72	85.3
	SD	0	4.6	4.6	0	0	4.6	4.6	9.2	9.2	0	0	4.6
34 °C	h_min_	0	16	24	56	64	74.7	0	24	42.7	52	61.3	72
	SD	0	0	0	0	0	4.6	0	0	4.6	5.7	4.6	0
	h_max_	8	18.7	50.7	56	66.7	77.3	18.7	34.7	48	56	64	77.3
	SD	0	4.6	4.6	0	4.6	4.6	4.6	4.6	8	0	0	4.6

**Table 2 insects-16-00480-t002:** Minimum and maximum duration of morphological changes in the thorax and mouthparts of *C. nigripes* after pupariation at different temperatures.

Temperature		Thorax (h)							Mouthparts (h)							
		I	II	III	IV	V	VI	VII	-	I	II	III	IV	V	VI	VII
19 °C	h_min_	0	56	93.3	125.3	141.3	144	176	0	37.3	53.3	64	85.3	133.3	162.7	181.3
	SD	0	0	4.6	4.6	4.6	0	8	0	4.6	4.6	8	12.2	9.2	4.6	4.6
	h_max_	56	85.3	122.7	136	149.3	168	192	32	50.7	56	77.3	130.7	154.7	173.3	192
	SD	8	4.6	9.2	0	4.6	8	0	0	12.2	8	12.2	4.6	4.6	4.6	0
22 °C	h_min_	0	34.7	66.7	98.7	120	122.7	136	0	32	44	50.7	72	114.7	122.7	146.7
	SD	0	23.1	4.6	12.2	0	4.6	0	0	0	5.7	4.6	0	4.6	4.6	4.6
	h_max_	42.7	58.7	93.3	112	120	125.3	149.3	24	40	44	64	106.7	117.3	138.7	149.3
	SD	4.6	4.6	9.2	0	0	4.6	4.6	0	8	5.7	0	4.6	4.6	4.6	4.6
25 °C	h_min_	0	34.7	74.7	85.3	88	93.3	109.3	0	24	32	36	56	80	93.3	106.7
	SD	0	4.6	4.6	4.6	0	4.6	12.2	0	0	0	5.7	21.2	13.9	4.6	12.2
	h_max_	26.7	66.7	80	90.7	88	101.3	117.3	18.7	26.7	32	56	74.7	85.3	101.3	117.3
	SD	4.6	4.6	0	4.6	0	12.2	4.6	4.6	4.6	0	22.6	16.7	4.6	9.2	4.6
28 °C	h_min_	0	34.7	58.7	64	72	77.3	85.3	0	18.7	26.7	37.3	48	69.3	77.3	93.3
	SD	0	4.6	9.2	8	0	4.6	4.6	0	4.6	4.6	4.6	8	4.6	4.6	4.6
	h_max_	29.3	50.7	58.7	72	72	80	96	10.7	18.7	32	40	69.3	72	85.3	96
	SD	9.2	9.2	9.2	0	0	0	0	4.6	4.6	8	8	4.6	0	4.6	0
31 °C	h_min_	0	24	42.7	58.7	68	74.7	82.7	0	16	24	26.7	45.3	68	74.7	85.3
	SD	0	0	4.6	4.6	5.7	4.6	4.6	0	0	0	4.6	12.2	5.7	4.6	4.6
	h_max_	18.7	37.3	53.3	64	68	74.7	85.3	16	16	24	37.3	64	68	77.3	85.3
	SD	4.6	9.2	9.2	0	5.7	4.6	4.6	0	0	0	12.2	0	5.7	4.6	4.6
34 °C	h_min_	0	24	42.7	48	56	64	69.3	0	16	24	29.3	36	58.7	69.3	77.3
	SD	0	0	4.6	0	0	0	4.6	0	0	0	9.2	5.7	4.6	4.6	4.6
	h_max_	18.7	34.7	45.3	52	56	64	77.3	8	18.7	32	37.3	52	61.3	69.3	77.3
	SD	4.6	4.6	9.2	5.7	0	0	4.6	0	4.6	0	16.7	5.7	4.6	4.6	4.6

“-” stage absent.

**Table 3 insects-16-00480-t003:** Minimum and maximum duration of each intra-puparial developmental stage of the antennae, wings, and legs of *C. nigripes* at different temperatures.

Temperature		Antennae (h)									Wings (h)									Legs (h)								
		-	I	II	III	IV	V	VI	VII	VIII	-	I	II	III	IV	V	VI	VII	VIII	-	I	II	III	IV	V	VI	VII	VIII
19 °C	h_min_	0	53.3	64	77.3	117.3	133.3	146.7	160	176	0	18.7	32	37.3	56	72	130.7	146.7	189.3	0	18.7	32	37.3	58.7	117.3	133.3	154.7	173.3
	SD	0	4.6	8	4.6	4.6	4.6	4.6	8	0	0	4.6	0	4.6	0	8	4.6	4.6	4.6	0	4.6	0	4.6	4.6	12.2	4.6	9.2	4.6
	h_max_	50.7	56	72	114.7	130.7	146.7	154.7	168	192	10.7	24	34.7	56	69.3	130.7	149.3	181.3	192	10.7	24	32	56	117.3	128	152	165.3	192
	SD	12.2	8	8	12.2	4.6	4.6	4.6	0	0	4.6	0	4.6	8	9.2	4.6	4.6	4.6	0	4.6	0	0	8	12.2	0	0	4.6	0
22 °C	h_min_	0	48	53.3	69.3	96	114.7	120	128	138.7	0	8	16	37.3	48	69.3	114.7	125.3	149.3	0	8	16	32	48	88	109.3	125.3	136
	SD	0	0	4.6	4.6	8	4.6	0	8	9.2	0	0	0	4.6	0	16.7	4.6	9.2	4.6	0	0	0	0	0	21.2	12.2	4.6	0
	h_max_	42.7	48	61.3	93.3	112	114.7	124	130.7	149.3	0	8	29.3	45.3	61.3	109.3	120	141.3	149.3	0	8	24	42.7	82.7	101.3	117.3	128	149.3
	SD	4.6	0	4.6	12.2	0	4.6	5.7	9.2	4.6	0	0	4.6	4.6	16.7	4.6	8	4.6	4.6	0	0	0	4.6	16.7	12.2	4.6	0	4.6
25 °C	h_min_	0	32	40	50.7	85.3	88	96	98.7	112	0	8	16	26.7	34.7	61.3	88	93.3	112	0	8	16	26.7	34.7	82.7	88	96	104
	SD	0	0	0	4.6	4.6	0	0	4.6	8	0	0	0	4.6	4.6	16.7	0	4.6	8	0	0	0	4.6	4.6	4.6	0	0	0
	h_max_	26.7	32	45.3	77.3	85.3	88	96	104	117.3	0	8	18.7	26.7	53.3	85.3	88	104	117.3	0	8	18.7	26.7	77.3	82.7	90.7	96	114.7
	SD	4.6	0	4.6	4.6	4.6	0	0	8	4.6	0	0	4.6	4.6	16.7	4.6	0	8	4.6	0	0	4.6	4.6	4.6	4.6	4.6	0	4.6
28 °C	h_min_	0	34.7	48	44	61.3	66.7	72	77.3	90.7	0	8	16	18.7	34.7	50.7	68	77.3	96	0	8	16	18.7	34.7	64	72	77.3	90.7
	SD	0	4.6	11.3	5.7	4.6	4.6	0	4.6	4.6	0	0	0	4.6	4.6	9.2	5.7	4.6	9	0	0	0	4.6	4.6	0	0	4.6	4.6
	h_max_	29.3	37.3	48	52	64	72	72	82.7	96	2.7	8	16	29.3	50.7	69.3	72	88	96	0	8	16	29.3	58.7	66.7	72	82.7	96
	SD	9.2	9.2	11.3	5.7	0	0	0	4.6	0	4.6	0	0	9.2	9.2	4.6	0	0	0	0	0	0	9.2	4.6	4.6	0	4.6	0
31 °C	h_min_	0	24	32	40	56	64	72	72	82.7	0	8	16	16	24	42.7	72	72	85.3	0	8	16	16	24	50.7	64	72	74.7
	SD	0	0	0	0	0	0	0	0	4.6	0	0	0	0	0	4.6	0	0	4.6	0	0	0	0	0	9.2	11.3	0	4.6
	h_max_	18.7	24	37.3	48	64	64	76	72	85.3	5.3	8	16	18.7	37.3	64	72	77.3	85.3	2.7	8	16	18.7	42.7	58.7	68	72	85.3
	SD	4.6	0	9.2	0	0	0	5.7	0	4.6	4.6	0	0	4.6	9.2	0	0	4.6	4.6	4.6	0	0	4.6	9.2	9.2	5.7	0	4.6
34 °C	h_min_	0	24	29.3	37.3	48	56	64	68	74.7	0	8	8	16	24	40	58.7	61.3	77.3	0	8	-	16	24	48	56	61.3	72
	SD	0	0	9.2	9.2	0	0	0	5.7	4.6	0	0	0	0	0	0	4.6	4.6	4.6	0	0	-	0	0	0	0	4.6	0
	h_max_	18.7	32	29.3	48	56	56	64	68	77.3	0	8	8	18.7	32	53.3	58.7	69.3	77.3	0	8	-	18.7	42.7	53.3	56	64	77.3
	SD	4.6	0	9.2	8	0	0	0	5.7	4.6	0	0	0	4.6	0	4.6	4.6	4.6	4.6	0	0	-	4.6	4.6	4.6	0	0	4.6

“-” stage absent.

**Table 4 insects-16-00480-t004:** Identification of major characteristic peaks of ATR-FTIR spectra during the intra-puparial period of *C. nigripes*.

Wave Number (cm^−1^)	Infrared Band
1100~1000	C-O(H) or C-C vibrational and PO^2^ symmetric stretching
1180~1100	C-O(H) stretching vibration
1270~1230	Two C-O telescopic vibration absorption
1330–1277	Amide III
1420~1350	C=O vibration of COO- of free fatty acids, free amino acids and peptides
1580~1510	Amide II (N-H bending coupled to C-N stretching)
1680~1610	Amide I (C=O stretching)
1760~1730	Lipids (C=O stretching vibration)

**Table 5 insects-16-00480-t005:** Accuracy (A), precision (P), recall (R), and F1 scores (F1-Score) of the PLS-DA and RF cross-validation of all spectral data of *C. nigripes* puparium at three different constant developmental temperatures (19 °C, 25 °C, and 31 °C).

	Accuracy (%)	19 °C		25 °C		31 °C	
Group		PLS-DA	RF	PLS-DA	RF	PLS-DA	RF
A		58.3%	63.3%	77.8%	41.7%	100%	89%
P		0.67	0.61	0.81	0.46	1	0.92
R		0.58	0.60	0.78	0.39	1	0.88
F1-Score		0.62	0.60	0.79	0.42	1	0.90

## Data Availability

The original contributions presented in this study are included in the article/Appendix A. Further inquiries can be directed to the corresponding author.

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
