# Peer review of "Estimating the Intra-Puparial Period of Chrysomya nigripes Aubertin Using Morphology and Attenuated Total Reflection Fourier Transform Infrared (ATR-FTIR) Spectroscopy"

_insects, 2025, doi:10.3390/insects16050480_

Round 1

Reviewer 1 Report

Comments and Suggestions for Authors

Chrysomya nigripes is a blow fly species that appears on highly decomposed corpses. It holds potential value for estimating the PMImin of corpses in the advanced decomposition stage or even the skeletonization stage. This manuscript attempts to use two methods, morphology and ATR – FTIR, to provide indicators for estimating the intra - puparial age of this species, which is useful in forensic investigations for determining the PMI and assisting in criminal cases.

Although the manuscript shows potential for publication, the following modifications are required:

  1. Introduction

Line 61: It is recommended to change "larval period" to "larval stage".

Lines 71 - 79: The authors cited two important references. However, these two are rather dated. The latest research advancements should be briefly mentioned. This will enable readers to better understand the current state of knowledge in this area.

Lines 80 - 86: The imaging techniques for intra-puparial period observation should be introduced first. Since imaging techniques essentially fall within the category of morphological methods, they should be expounded upon after the introduction of the morphological studies of the intra - puparial period.

Lines 98 - 100: As far as I know, there have already been several published researches on the application of ATR-FTIR in the developmental studies of forensic entomology. The authors are kindly requested to revise this sentence. Additionally, it would be appreciated if the authors could also provide some backgrounds regarding ATR - FTIR combined with chemometric methods.

  1. Materials and Methods

Lines 113 - 114: "Laboratory population establishment of C. nigripes was carried out according to Guo et al. [26]." Although readers can obtain information about the population establishment from the previous research, it is evident that this is more difficult compared to getting it directly from the present study. Therefore, the reviewer suggests that the authors briefly describe the method of population establishment.

Line 122: It is recommended to change "scalded" to "killed".

Line 124: "pupae" should be revised as "puparium".

Lines 130 - 133: The authors used ultrasonic cleaning for the puparium. Were these cleaned puparium directly subjected to testing using the FTIR spectrometer? Could the moisture within the puparia potentially impact the experimental results? Considering that moisture might interfere with the FTIR analysis, it is crucial to clarify the handling process of the puparium.

  1. Results and Discussion

Line 222: "the respiratory horns darkened in color and located in the anterior part of the pupae". This characteristic does not seem to be exclusive to sub - stage C. In fact, it should first appear in sub - stage B. Please make the necessary revisions.

Lines 164 - 173: The authors are advised to consider adding a table or a figure to more clearly present the comparison of the intra-puparial development times of this species across different studies.

Line 296: "pupal" should be revised as "intra-puparial".

Starting from Table 3, the subsequent row numbers are not consecutive with the ones before it. When the authors revise the manuscript, please make sure to avoid this discrepancy. For the issues occurring after Table 3, the reviewers have specified them according to the new row numbers.

Line 1: "Aldrichina grahami" should be revised as "Calliphora grahami".

Lines 54 - 56: In the experimental method regarding the FTIR spectrometer, the authors adopted a sampling interval of 24 hours. However, it is necessary to clarify why a 48-hour interval was used during the PCA and what the basis for this choice is.

Lines 61 - 67: The authors employed two machine learning models, namely PLS-DA and RF. It is required that the authors provide a reasonable explanation and conduct a discussion on why these two specific models were chosen and why other models were not considered.

Table 5: The header of this table is not displayed properly. Please make the necessary modifications.

References

There should be a space before the square brackets of the references in the main text. Please make this correction throughout the entire text. Additionally, please conduct a thorough check of the reference list. I have identified some formatting issues, so please carefully verify and rectify them in accordance with the Instructions for Authors.

Author Response

We feel great thanks for your professional review work and recognition on our article, these comments are all valuable and helpful for improving our article. In this revised version, changes to our manuscript were all highlighted within the document by using red colored text.

Reviewer #1:

1.Introduction

Line 61: It is recommended to change "larval period" to "larval stage".

Response: Thanks for your corrections. We have replaced " larval period " with " larval stage ". The word appears again in lines 61 and 70, both of which have been amended.

Lines 71 - 79: The authors cited two important references. However, these two are rather dated. The latest research advancements should be briefly mentioned. This will enable readers to better understand the current state of knowledge in this area.

Response: Thanks for your suggestion. Because it's two iconic studies, one established a nomenclature for the different stages of the intra-puparial development,another was the first species-level forensic study to directly link developmental periods to morphological changes. It is the origin of the morphological changes in the different stages of the intra-puparial development and the naming of the stages. This is also used as a standard reference in new related articles. In accordance with your suggestion, we have incorporated the recent relevant researches on the intra-puparial development of forensically important insects. The specific modifications are as follows: These criteria are now gradually being refined and applied to the study of the intra-puparial morphology of various fly species [12-14].

Lines 80 - 86: The imaging techniques for intra-puparial period observation should be introduced first. Since imaging techniques essentially fall within the category of morphological methods, they should be expounded upon after the introduction of the morphological studies of the intra - puparial period.

Response: Thanks for your corrections. We have placed the introduction of imaging techniques in the first part of this paragraph as part of morphology, before introducing the ever-increasing number of new research methods. The specific detail is as follows: In addition to studies removing the puparium to observe morphological changes within the pupae under stereomicroscope, recent researches In recent years, researchers have also begun to use a variety of imaging techniques for intra-puparial stage observa-tion, including optical coherence tomography, microcomputed tomography, and scanning electron microscopy. In addition to studies removing the puparium to observe morphological changes within the pupae, research on estimating the age of fly pupae is growing.Meanwhile, Several studies have established methods for estimating intra-puparial age using gene expression changes , cuticular hydrocarbons and gas/liquid chromatography-mass spectrometry .

Lines 98 - 100: As far as I know, there have already been several published researches on the application of ATR-FTIR in the developmental studies of forensic entomology. The authors are kindly requested to revise this sentence. Additionally, it would be appreciated if the authors could also provide some backgrounds regarding ATR - FTIR combined with chemometric methods.

Response: Thanks for your suggestion. We have revised this sentence. The specific detail is as follows: The combination of ATR-FTIR and chemometrics provides a powerful tool for chemical analysis, enabling high quality quantitative and qualitative results to be obtained along-side rapid, non-destructive analysis. The application of this method is promising, especially in research areas where high throughput and sensitivity are required. However, However, compared to the use of ATR-FTIR in other areas of forensic science, the appli-cation of ATR-FTIR in forensic entomology, especially for the aging of sarcosaproph-agous insects, is rare [26,27].

2.Materials and Methods

Lines 113 - 114: "Laboratory population establishment of C. nigripes was carried out according to Guo et al. [26]." Although readers can obtain information about the population establishment from the previous research, it is evident that this is more difficult compared to getting it directly from the present study. Therefore, the reviewer suggests that the authors briefly describe the method of population establishment.

Response: Thanks for your suggestion. We've made corrections. The specific detail is as follows: The wandering larvae were collected in Zhongshan, China (22°30′ N, 113°23′ E) and transported to the forensic entomology laboratory of Soochow University for the establishment of laboratory population. The larvae were reared to adults in an intelligent light incubator at 25 °C. After eclosion, the adults were maintained in a rearing cage measuring 50 × 50 × 50 cm, where an equal mixture of powdered milk and white sugar was provided to promote sexual maturation of the adults. Following the maturation of the adults, 40 g of decomposed pork was placed in a beaker covered with a layer of pig skin, and the beaker was placed in a rearing cage to attract eggs laying. The eggs were observed every hour, and after laying, the eggs were transferred to an intelligent light incubator. This procedure was repeated until the population grew to include around 2500 adults in each rearing cage.

Line 122: It is recommended to change "scalded" to "killed".

Response: Thanks for your corrections. We have replaced " scalded " with " killed ". The specific detail is as follows: The samples were killed with hot water (> 90°C) and placed in 80% ethanol for storage.

Line 124: "pupae" should be revised as "puparium".

Response: Thanks for your corrections. We have replaced " pupae " with " puparium ". The specific detail is as follows: Then, the puparium was carefully removed with an insect needle and tweezers, and the intra-puparial morphology was observed, recorded and photographed under a Nexcope NSZ818 research-grade compound achromatic parallel-light stereomicroscope.

Lines 130 - 133: The authors used ultrasonic cleaning for the puparium. Were these cleaned puparium directly subjected to testing using the FTIR spectrometer? Could the moisture within the puparia potentially impact the experimental results? Considering that moisture might interfere with the FTIR analysis, it is crucial to clarify the handling process of the puparium.

Response: Thanks for your corrections. This sentence was incorrectly expressed and has been modified. The specific detail is as follows: Subsequently, all puparia pieces were placed on a dry piece of paper towel to remove any excess moisture. After that, the puparium was left to air-dry in a well-ventilated environment for 1 h. Once thoroughly dried, the samples were then carefully placed in an 1.5 ml EP tube, and stored at -80°C for further processing. All the samples stored at -80 °C will undergo ATR-FTIR detection within a week.

3. Results and Discussion

Line 222: "the respiratory horns darkened in color and located in the anterior part of the pupae". This characteristic does not seem to be exclusive to sub - stage C. In fact, it should first appear in sub - stage B. Please make the necessary revisions.

Response: Thanks for your corrections. We've moved this feature to the sub – stage B.

Lines 164 - 173: The authors are advised to consider adding a table or a figure to more clearly present the comparison of the intra-puparial development times of this species across different studies.

Response: Thanks for your suggestion. The only articles on intra-puparial period have been published by Guo et al. and Li et al. and O'Flynn, of which O'Flynn only observed the intra-puparial period at one temperature, so there are fewer articles on this study, and fewer comparable data, so no graphs have been placed for comparisons.

Line 296: "pupal" should be revised as "intra-puparial".

Response: Thanks for your corrections. We have replaced " pupal " with " intra-puparial ". The specific detail is as follows: The overall intra-puparial morphology of C. nigripes is described above, but its details are not always clearly recognizable, especially when the insect evidence collected at the scene of the crime is incomplete.

Starting from Table 3, the subsequent row numbers are not consecutive with the ones before it. When the authors revise the manuscript, please make sure to avoid this discrepancy. For the issues occurring after Table 3, the reviewers have specified them according to the new row numbers.

Response: Thanks for your corrections. We've made changes to the line numbers.

Line 1: "Aldrichina grahami" should be revised as "Calliphora grahami".

Response: Thanks for your corrections. We have replaced " Aldrichina grahami " with " Calliphora grahami ". The specific detail is as follows: The methodology of this study is consistent with that of the methods of Brown et al., Wang et al. and Li et al., and the changes observed in various characteristic organs within the puparium of C. nigripes were similar to those of Calliphora vicina, Lucilia illustris, Calliphora grahami, and Sarcophaga peregrina. The above results suggest that the subdivision of morphological characteristics of the whole, parts and organs within the puparium over time provides a reliable basis for age estimation during the intra-puparial stage.

Lines 54 - 56: In the experimental method regarding the FTIR spectrometer, the authors adopted a sampling interval of 24 hours. However, it is necessary to clarify why a 48-hour interval was used during the PCA and what the basis for this choice is.

Response: Thank you for your question. We used 24-hour intervals samples for our spectroscopic experiments, and all of these samples were used for the chemometric analyses, except that using 24-hour intervals for PCA mapping was too much data, as the 19°C sampling time amounted to 216 hours, and the grouping was not ideal. Therefore adopted 48 hours interval graphing, the amount of data is slightly smaller, grouping is more obvious and regular.

Lines 61 - 67: The authors employed two machine learning models, namely PLS-DA and RF. It is required that the authors provide a reasonable explanation and conduct a discussion on why these two specific models were chosen and why other models were not considered.

Response: Thank you for your question. We have added the appropriate discussion in the text, the specific detail is as follows: In chemometrics, selecting an appropriate machine learning model is pivotal for ensuring the accuracy and reliability of data analysis. The PLS-DA employed in this study is well-suited for handling high-dimensional data, particularly in spectral data analysis. It can elucidate the relationship between variables and classification results, which is essential for understanding the connection between components and their spectra. Conversely, RF can evaluate the contribution of individual features to classification outcomes, aiding in identifying the key variables in the analysis. Although many other machine learning models, such as support vector machines and neural networks, demonstrate good performance in certain scenarios, they generally demand more parameter tuning and intricate model design. These complex models also require longer training and prediction times. In contrast, PLS-DA and RF are computationally more efficient, making them more appropriate for rapid analysis and real-time applications in this study. However, future research should still assess the data prediction accuracy of different models to ensure more precise PMImin.

Table 5: The header of this table is not displayed properly. Please make the necessary modifications.

Response: Thanks for your corrections. We've made changes to the header of the table.

References

There should be a space before the square brackets of the references in the main text. Please make this correction throughout the entire text. Additionally, please conduct a thorough check of the reference list. I have identified some formatting issues, so please carefully verify and rectify them in accordance with the Instructions for Authors.

Response: Thanks for your corrections. We have made changes in the text and corrected the formatting of references.

Reviewer 2 Report

Comments and Suggestions for Authors A lovely piece of work with significant value for the field. A few clarifications needed, as listed below (minor):   Abstract It would be helpful at this stage to know that a segment of the puparium (puparial case) is used for the ATR-FTIR, not the whole pupa as it reads (to me) at present.    Introduction Line 94 mentions simple preservation - this is hugely important for CSIs so referencing our standards and guidelines (Amendt et al 2007) would be valuable to add here.    Methods Line 121 - remove 'and' before 'five pupae'. Line 122 - pupae are killed rather than scalded Line 123 - ethanol, not alcohol Line 130 - clarify that these samples (for ATR-FTIR) were preserved the same as the pupae for morphological analysis Line 131 - how long was it between sampling, preservation and puparium cutting for ATR-FTIR? Does it have to occur immediately upon collection, or can pupae be preserved for a week (for example) before creating this sample? This will have bearings on its use in forensic investigation.  Line 135 - why are samples needed to be preserved at -80; will -20 not suffice? This may be a limiting factor for use of this method and should be explained/discussed.    Results Three repeats of the experiment were conducted at each temperature, resulting in 15 pupae per timepoint. Was any work conducted/observations made regarding the intra- vs inter-replicate differences? Or was each repeat similar enough to bring together to give the mean/max/min ages for each stage? Fig 2 - there is a distinct 'bubble' on each graph between stages D and H, where the range between max and min is at its greatest. I dont recall this being highlighted or discussed as to why this might be, and its implications for PMImin estimation using samples in this age range.  Line 239 - some wings are folded should be changed to 'partially folded'.  Line 260 - black is a solid tone, it's neither light or dark. Remove 'dark'.  Figure 5 - These are images of the thorax, which are the same as Figure 6. Abdominal images needed. The legend refers to 'ventral' bristles, but its often the dorsal bristles referred to. Please check this.  Line 39 - how easy is it to cut such a small piece of the puparium, when it is so brittle? Any diagrams/figures to advise where and how? Fig 11 19oC shows a split in the waveforms, with 192-216h being separated from the others - this should be discussed. It is not shown at the other temperatures.  Line 54 - clustering circles on the PCA plots would make points raised here clearer to see. What about the trends here, as I think clear groups can be noted?   Discussion Line 42 in the results comments on the analysis of pupae using both morphological and ATR-FTIR methods which is excellent, but this was not continued through to the PCA/Chemometrics. I feel there is a missed opportunity here to combine both datasets (yes, quantification of morphology will be required) and examine using machine learning whether PMImin estimation is bolstered using both techniques together. 

Author Response

We feel great thanks for your professional review work and recognition on our article, these comments are valuable and helpful for improving our article. In this revised version, changes to our manuscript were all highlighted within the document by using red colored text.

Reviewer #2:

Abstract

 It would be helpful at this stage to know that a segment of the puparium (puparial case) is used for the ATR-FTIR, not the whole pupa as it reads (to me) at present. 

Response: Thanks for your suggestion. We've made a correction. The specific detail is as follows: The spectral data within the wavenumber range of 1800 - 900 cm⁻¹, collected from the second thoracic segment of all puparia, were processed. Through this procedure, the mean values of ATR-FTIR spectra of C. nigripes of puparia at each intra-puparial age un-der various constant temperature conditions were obtained.

Introduction

Line 94 mentions simple preservation - this is hugely important for CSIs so referencing our standards and guidelines (Amendt et al 2007) would be valuable to add here. 

Response: Thanks for your suggestion. Recognizing the critical significance of simple preservation for crime scene investigations (CSIs), as you pointed out, we have now incorporated the seminal work by Amendt et al. (2007) as a reference.

Methods

 Line 121 - remove 'and' before 'five pupae'.

Response: Thanks for your corrections. We’ve removed “and” before “five pupae”. The specific detail is as follows: When about 50% of the wandering larvae formed white pre-pupae, five pupae were sampled every 8h until eclosion at each temperature.

Line 122 - pupae are killed rather than scalded

Response: Thanks for your corrections. We have replaced " scalded " with " killed ". The specific detail is as follows: The samples were killed with hot water (> 90°C) and placed in 80% ethanol for storage.

Line 123 - ethanol, not alcohol

Response: Thanks for your corrections. We have replaced " alcohol " with " ethanol ". The specific detail is as follows: The samples were killed with hot water (> 90°C) and placed in 80% ethanol for storage.

Line 130 - clarify that these samples (for ATR-FTIR) were preserved the same as the pupae for morphological analysis

Response: Thank you very much for your valuable suggestion. I surmise that what you originally meant was that we adopted the same rearing method as that used in the morphological analysis. To address this, we have incorporated the statement: " The pupae for ATR-FTIR detection were reared at 19°C, 25°C and 31°C in the same way as the samples for morphological analysis." Nonetheless, there are notable differences between the ATR-FTIR analysis and the morphological study with respect to the preservation of the pupae. The pupae utilized for morphological analysis were preserved in 80% ethanol to facilitate subsequent observations. In contrast, for the puparium intended for spectroscopic analysis, they were washed and cleaned promptly right after sampling. Subsequently, the puparium was removed and stored at -80°C.

Line 131 - how long was it between sampling, preservation and puparium cutting for ATR-FTIR? Does it have to occur immediately upon collection, or can pupae be preserved for a week (for example) before creating this sample? This will have bearings on its use in forensic investigation. 

Response: Thank you very much for your valuable suggestion. In this study, the pupae were cleaned right after sampling, and subsequently, the puparium was carefully peeled off. Second thoracic segment of all puparia was carefully excised. After drying the puparia pieces in a well-ventilated environment for one hour, they were placed into EP tubes for storage at -80°C. All the samples stored at -80°C will undergo ATR-FTIR detection within a week. We believe that the most optimal approach is to conduct ATR-FTIR detection immediately after sampling. However, the reason why the samples were stored at -80°C in this study is that the usage of the ATR-FTIR instrument requires prior reservation. Therefore, we tend to accumulate a certain number of samples and perform the measurement in one batch. In the context of forensic investigations, it is highly advisable to process the samples immediately after they are collected. The specific detail is as follows: The pupae for ATR-FTIR detection were reared at 19°C, 25°C and 31°C in the same way as the samples for morphological analysis. Five pupae were taken from each temperature at 24-h intervals until eclosion, and the last samples were five empty puparia. After sampling, the surface of the pupae/puparia were cleaned with an ultrasonic cleaner (Fan Ying Technology Co. Ltd., Zhongshan, China),). The dorsal puparium at the second thoracic segment (ca. 1.1 × 2.2 mm) of each pupa was removed with forceps and scissors. Subsequently, all puparia pieces were placed on a dry piece of paper towel to remove any excess moisture. After that, the puparium was left to air-dry in a well-ventilated environment for 1 h. Once thoroughly dried, the samples were then carefully placed in an 1.5 ml EP tube, and stored at -80°C for further processing. All the samples stored at -80 °C will undergo ATR-FTIR detection within a week.

Line 135 - why are samples needed to be preserved at -80; will -20 not suffice? This may be a limiting factor for use of this method and should be explained/discussed. 

Response: Thanks for your suggestion. At -20°C, biomolecules such as proteins and nucleic acids in biological samples may still be degraded and deformed, whereas at -80°C the rate of such degradation is significantly reduced which helps to maintain the integrity of the samples, especially since our spectroscopic experiments are closely related to these biomolecules. As we mentioned in our response to the previous question, in actual criminal cases, it is feasible to conduct the detection of samples right after they are collected. Certainly, considering that the degradation of biomolecules is highly unlikely to occur at -80°C, we firmly believe that the results of this study are reliable and robust.

 Results 

Three repeats of the experiment were conducted at each temperature, resulting in 15 pupae per timepoint. Was any work conducted/observations made regarding the intra- vs inter-replicate differences? Or was each repeat similar enough to bring together to give the mean/max/min ages for each stage?

Response: Thanks for your question. We took five pupae at 8-h intervals and repeated each temperature three times. As you pointed out, there are indeed differences between different intra-replicates and inter-replicates. In Figure 2 and Tables 1 to 3, we presented these differences by showing the average minimum value±SD and the average maximum value±SD. Please review our figures and tables again.

Fig 2 - there is a distinct 'bubble' on each graph between stages D and H, where the range between max and min is at its greatest. I dont recall this being highlighted or discussed as to why this might be, and its implications for PMImin estimation using samples in this age range. 

Response: Thanks for your question. We have added a paragraph to discuss this issue. The details are as follows: Similar to other species of Diptera, the development within the puparium of the C. nigripes goes through the pre-pupal, cryptocephalic pupal, phanerocephalic pupal, and pharate adult stages. However, the entire process of intra-puparial development is not linear. It is rapid in the early stages (e.g., stages A-C in Fig. 2), slow during the middle stages (stages D-H in Fig. 2), and becomes rapid again in the later stages (stages I-L in Fig. 2). As a result, a distinct 'bubble' is formed on each graph in Figure 2. In fact, this represents a noticeable limitation of this method. In actual cases, the investigators can only use the minimum value as a reference for PMImin. This is precisely why many current studies, including the present study, combine multiple methods for estimating the intra-puparial age.

Line 239 – ‘some wings are folded’ should be changed to 'partially folded'. 

Response: Thanks for your corrections. We have replaced " some wings are folded " with " partially folded ". The specific detail is as follows: wing veins are visible, and partially folded.

Line 260 - black is a solid tone, it's neither light or dark. Remove 'dark'. 

Response: Thanks for your corrections. We have removed ‘dark’. The specific detail is as follows: wings black.

Figure 5 - These are images of the thorax, which are the same as Figure 6. Abdominal images needed. The legend refers to 'ventral' bristles, but its often the dorsal bristles referred to. Please check this. 

Response: Thanks for your corrections. We've corrected the Figure 5. We have replaced " ventral seta" with " hairs ".

 Line 39 - how easy is it to cut such a small piece of the puparium, when it is so brittle? Any diagrams/figures to advise where and how?

Response: Thanks for your question. Regarding the concern about cutting a small piece of the puparium, I would like to clarify our sampling approach. In our study, we first cleaned the pupae and subsequently removed the puparium. During this process, the pupae remained alive, and the puparium was not brittle. Once the technique was proficiently mastered, the entire puparium could be removed intact. Subsequently, we carried out the cutting of specific sections of the puparium. We consistently targeted the same area on each pupa to ensure the control of variables, as the chemical composition of the puparium may vary across different regions. This approach helped us maintain consistency and reliability in our experimental results.

Fig 11 19°C shows a split in the waveforms, with 192-216h being separated from the others - this should be discussed. It is not shown at the other temperatures. 

Response: Thanks for your question. This may due to the chemical reaction. The details are as follows: The 192h and 216h average spectra at 19°C are separated from the other time spectra, this is because of the long developmental duration of this species at 19°C, with eclosion ranging from 192-216 h. The samples may undergo chemical reactions (e.g. oxidation, degradation, polymerisation) over a long period of time (192h, 216h) to produce sub-stances containing strong absorptive functional groups (e.g. hydroxyl -OH, carbonyl C=O). These functional groups increase the absorption of infrared light at specific wave-lengths, resulting in a significant decrease in reflectance.

Line 54 - clustering circles on the PCA plots would make points raised here clearer to see. What about the trends here, as I think clear groups can be noted?   

Response: Thanks for your corrections. We have modified the picture and the explanation as follows: The results of the PCA are shown in Fig 12. From the infrared spectral score plots of the samples, it can be seen that, at different constant development temperatures (19°C, 25°C, and 31°C), PC1 explains 55.00%, 52.53%; and 53.03% of the centralized variance, and PC2 explains 27.09%, 28.68% and 15.31% of the centralized variance, respectively. The interpretation of variation of the score plots of the spectral data at different times (d) on PC1 and PC2 were 82.09%, 81.21%, and 68.34%. Although the samples at different time points at different temperatures had inter-group reproducibility and intergroup differentiation, there will still be intersections. The longer the time intervals, the more pronounced the distinction between the groups. There was a clear separation trend between the 0-48h and 192-216h groups at 19°C, 0-48h and 96-144h at 25°C. Therefore, we combined PLS-DA and RF to further analysis these data.

Discussion

 Line 42 in the results comments on the analysis of pupae using both morphological and ATR-FTIR methods which is excellent, but this was not continued through to the PCA/Chemometrics. I feel there is a missed opportunity here to combine both datasets (yes, quantification of morphology will be required) and examine using machine learning whether PMImin estimation is bolstered using both techniques together. 

Response: Thanks for your question. We believe that there are two main advantages to combining these two methods. Firstly, calculate PMImin separately with each method, and then use the intersection of the results from the two methods to further narrow down the PMImin time window. This way, we can take full advantage of the unique aspects of each method. Secondly, there are situations where the ATR-FTIR device may not be easily accessible. In such cases, we can estimate PMImin by observing the internal structure within the puparium. Also, when peeling the puparium, inexperienced investigators might potentially damage the internal structure of fly pupae. In those circumstances, we can resort to the combination of ATR-FTIR and chemometrics to make up for any possible morphological data loss.

We have already elaborated on this issue in the conclusion, as follows: The intra-puparial period accounts for 50% of the entire immature stage, which is of great significance for the estimation of PMImin. This study proposes that using ATR-FTIR in combination with chemometrics serves as a supplementary tool for morphological aging assessment. There are two main advantages to combining these two methods. Firstly, calculate PMImin separately with each method, and then use the intersection of the results from the two methods to further narrow down the PMImin time window. This way, we can take full advantage of the unique aspects of each method. Secondly, there are situations where the ATR-FTIR device may not be easily accessible. In such cases, investigators can estimate PMImin by observing the internal structure within the puparium. Also, when peeling the puparium, inexperienced investigators might potentially damage the internal structure of fly pupae. In those circumstances, the combination of ATR-FTIR and chemometrics can be used to make up for potential losses of morphological data. Overall, this study presents a novel approach for estimating the age of forensically important C. nigripes pupae, and consequently, it helps in accurately estimating the PMImin for forensic entomology investigations.